# Genetic Susceptibility in Endothelial Injury Syndromes after Hematopoietic Cell Transplantation and Other Cellular Therapies: Climbing a Steep Hill

Paschalis Evangelidis [1], Nikolaos Evangelidis [1], Panagiotis Kalmoukos [1], Maria Kourti [2], Athanasios Tragiannidis [3] and Eleni Gavriilaki [1,*]

[1] 2nd Propedeutic Department of Internal Medicine, Hippocration Hospital, Aristotle University of Thessaloniki, 54642 Thessaloniki, Greece; pascevan@auth.gr (P.E.); evangeln@auth.gr (N.E.); kalmoukosp@yahoo.gr (P.K.)

[2] 3rd Department of Pediatrics, Hippocration Hospital, Aristotle University of Thessaloniki, 54642 Thessaloniki, Greece; makourti@auth.gr

[3] 2nd Department of Pediatrics, AHEPA University Hospital, Aristotle University of Thessaloniki, 54636 Thessaloniki, Greece; atragian@auth.gr

* Correspondence: gavriiel@auth.gr; Tel.: +30-697-3841-671

**Abstract:** Hematopoietic stem cell transplantation (HSCT) remains a cornerstone in the management of patients with hematological malignancies. Endothelial injury syndromes, such as HSCT-associated thrombotic microangiopathy (HSCT-TMA), veno-occlusive disease/sinusoidal obstruction syndrome (SOS/VOD), and capillary leak syndrome (CLS), constitute complications after HSCT. Moreover, endothelial damage is prevalent after immunotherapy with chimeric antigen receptor-T (CAR-T) and can be manifested with cytokine release syndrome (CRS) or immune effector cell-associated neurotoxicity syndrome (ICANS). Our literature review aims to investigate the genetic susceptibility in endothelial injury syndromes after HSCT and CAR-T cell therapy. Variations in complement pathway- and endothelial function-related genes have been associated with the development of HSCT-TMA. In these genes, *CFHR5*, *CFHR1*, *CFHR3*, *CFI*, *ADAMTS13*, *CFB*, *C3*, *C4*, *C5*, and *MASP1* are included. Thus, patients with these variations might have a predisposition to complement activation, which is also exaggerated by other factors (such as acute graft-versus-host disease, infections, and calcineurin inhibitors). Few studies have examined the genetic susceptibility to SOS/VOD syndrome, and the implicated genes include *CFH, methylenetetrahydrofolate reductase*, and *heparinase*. Finally, specific mutations have been associated with the onset of CRS (*PFKFB4*, *CX3CR1*) and ICANS (*PPM1D*, *DNMT3A*, *TE2*, *ASXL1*). More research is essential in this field to achieve better outcomes for our patients.

**Keywords:** allogeneic; autologous; CAR-T; CRS; endothelial; hematopoietic stem cell transplantation; HSCT-TMA; ICANS; gene; SOS/VOD

## 1. Introduction

Hematopoietic stem cell transplantation (HSCT) constitutes a cornerstone in the management of hematological malignancies, while several complications increase the morbidity and mortality in HSCT recipients [1,2]. Graft-versus-host disease (GVHD) is considered the major cause of death in these patients, mainly due to infections complicating this clinical entity, while cardiovascular disease (CVD) burden is also substantially increased, as shown by both clinical and laboratory data [3–6]. Endothelial dysfunction and injury plays a substantial role in the pathogenesis of vascular complications post-HSCT, while it is also implicated in the development of HSCT-associated thrombotic microangiopathy (HSCT-TMA), veno-occlusive disease/sinusoidal obstruction syndrome (SOS/VOD), capillary leak syndrome (CLS), and GVHD [7,8]. Markers of endothelial injury such as the

Endothelial Activation and Stress Index (EASIX) have been used as predictors of survival in allogeneic HSCT (alloHSCT) survivors [9,10]. Genetic susceptibility has been recognized as a predisposing risk factor in the development of endothelial injury syndromes [11,12].

Chimeric antigen receptor-T (CAR-T) immunotherapies have become the standard of care in the treatment approach of patients with relapsed/refractory b-cell malignancies [13]. To date, two CAR-T cell therapies have been approved for the treatment of relapsed/refractory multiple myeloma: idecabtagene vicleucel and ciltacabtagene autoleucel [14]. Furthermore, lisocabtagene maraleucel, tisagenlecleucel, brexucabtagene autoleucel, and axicabtagene ciloleucel are administered in patients with relapsed/refractory lymphomas and b-cell acute lymphoblastic leukemia [15]. Cytopenias and subsequent infections might complicate patients who receive CAR-T cell immunotherapy [16–18]. Toxicities, which include cytokine release syndrome (CRS) and immune effector cell-associated neurotoxicity syndrome (ICANS), limit the therapeutic efficacy of this treatment approach [19–21]. CRS might also complicate patients with severe coronavirus disease (COVID-19), influenza, and those who receive immunotherapy for solid tumors [22,23]. CRS and ICANS can be described as "endotheliopathies" because endothelial dysfunction plays a significant role in the pathogenesis of these syndromes [24,25]. Genetic factors have been associated with the development of CRS post-CAR-T infusion [26].

The aim of the current literature review was to investigate the genetic susceptibility to the manifestation of endothelial injury syndromes (HSCT-TMA, SOS/VOD) post-HSCT. Moreover, we summarized the key gene mutations' predispositions to the development of CRS after CAR-T cell immunotherapy. For this purpose, a literature review was performed using keywords such as "endothelial injury", "transplant-associated thrombotic microangiopathy", "SOS/VOD", "gene", "genetic", "HSCT", "allogeneic", "CAR-T", and "CRS", in different combinations, to identify relevant studies published in the English language. In the era of precision and personalized medicine, a better understanding of these complex disease entities is crucial not only for better outcomes for our patients but also for the early prevention of the toxicities following treatment.

## 2. HSCT-TMA

### 2.1. HSCT-TMA: Complement Dysregulation and Endothelial Dysfunction in the Spotlight

HSCT-TMA is a distinct type of thrombotic microangiopathy, following HSCT, and is characterized by the clinical triad of thrombocytopenia, macroangiopathic hemolytic anemia, and target-organ damage [27]. In a meta-analysis published by Van Benschoten et al., the pooled incidence of HSCT-TMA after alloHSCT was 12% (95% confidence interval, range 9%–16%) [28]. HSCT-TMA can complicate both allogeneic and autologous HSCT, mainly affecting pediatric patients with neuroblastoma, but it is more prevalent in alloHSCT [29]. A three-hit hypothesis has been described for the pathogenesis of HSCT-TMA [12]. The first hit includes underlying predisposing factors: female sex, African American race, non-malignant hematological disorder, history of previous HSCT, and genetic variants [7,28]. The second hit of transplant-related risk factors, such as transplant conditioning regimen-related toxicity, total-body irradiation, unrelated donor transplantation, and human leukocyte antigen (HLA) mismatch, leads to endothelial injury, and a procoagulant state is developed [30,31]. Balassa et al. showed that the presence of HLA-DRB1*11 antigen ($p = 0.034$) was associated with the development of HSCT-TMA [32]. Finally, post-HSCT risk factors, including the administration of calcineurin (CNI) or mTOR inhibitors, the development of acute GVHD (aGVHD), and infections, result in complement activation [33,34]. The activation of classical, lectin, and alternative pathways of the complement is implemented, leading to the formulation of membrane attack complex (MAC) (C5b-9) and subsequent complement-mediated cell lysis [35,36]. The pathophysiology of HSCT-TMA is similar in pediatric and adult HSCT recipients.

The clinical implications of the syndrome are the result of organ injury and include manifestations from the kidneys (hypertension, proteinuria, acute kidney injury), gastrointestinal (GI) tract, central nervous system (headache, confusion, seizures, posterior

reversible encephalopathy), and lungs (pulmonary arterial hypertension, pulmonary insufficiency) [37]. Moreover, multiple-organ failure might be established [38]. Various criteria have been proposed for the diagnosis of patients with TA-TMA, while in the most recently published criteria, HSCT-TMA is diagnosed based on clinical and laboratory data, and kidney/GI biopsy can be implemented but is not essential for the diagnosis [37]. No treatment guidelines are available for the management of HSCT-TMA. Historically, several treatment approaches have been investigated for HSCT-TMA, including the discontinuation of CNIs, therapeutic plasma exchange, defibrotide, immunosuppressant agents, and rituximab, with various outcomes for patients [39]. Eculizumab (C5 complement inhibitor) has been shown to be efficient and safe in HSCT-TMA [40,41]. However, an increase in the overall survival of patients who receive eculizumab compared to those who do not, has not yet been shown [42,43]. Next-generation complement therapeutics such as Pegcetacoplan, Narsoplimab, Coversin, and Ravulizumab are under investigation for HSCT-TMA in both pediatric and adult patients [44–48].

### 2.2. Genetic Susceptibility to HSCT-TMA

As mentioned above, genetic variants may play a role in the pathogenesis of HSCT-TMA. Jodele et al. performed targeted genomic analysis in 17 genes (*CFH*, *CFHR1*, *CFHR3*, *CFHR4*, *CFHR5*, *CD55*, *CD59*, *CD46*, *CFI*, *CFB*, *CFP*, *C5*, *ADAMTS13*, *CFD*, *C3*, *C4BPA*, and *THBD*) in 90 alloHSCT recipients (34 with HSCT-TMA) [49]. In total, 239 variants were identified in 17 genes in the initial control and 42 were considered likely functional variants. A total of 15 of them were associated before with the development of thrombotic microangiopathies (6 were reported to be pathogenic and 9 were of uncertain clinical significance). Pathogenic variants previously described in other thrombotic microangiopathies were shown in *CFHR5*, *CFI*, and *ADAMTS13* (a disintegrin and metalloproteinase with a thrombospondin type 1 motif, member 13) genes (only in patients with HSCT-TMA). ADAMTS13 is a metalloprotease enzyme that cleaves von Willebrand factor (vWF), a large protein involved in the formulation of blood clots [50]. Deficiency in the activity of plasma ADAMTS13 (<10%) is typical in thrombotic thrombocytopenic purpura (TTP) [51]. Despite the identification of mutations in the *ADAMTS13* gene, the activity of ADAMTS13 was between 43 and 50% in the whole group of patients, excluding the diagnosis of TTP [49,52]. *CFHR5* is part of a complement factor H (CFH) gene cluster, located on chromosome 1, encoding a protein that can bind to complement component 3 b (C3b), acting as an alternative complement pathway regulator [53]). Mutations on *CFHR5* genes have been connected with FHR5 nephropathy, a glomerulopathy, characterized by the accumulation of complement C3 deposits in the glomerulus [54]. Complement factor I (CFI) inactivates C3b and C4b complement factors, regulating both classical and alternative pathways, while loss-of-function mutations in the *CFI* gene lead to increased complement activation [55]. In total, 65% (22 of 34) of HSCT-TMA patients were found to have at least one gene variant in comparison to 9% (4 of 43) of those without TMA after the transplantation ($p < 0.0001$). The median number of gene variants found in HSCT-TMA patients was one and zero in those without ($p < 0.0001$). Gene variants were more common in nonwhite patients, compared to white patients ($p < 0.0001$). Moreover, patients with $\geq 3$ gene variants (all were nonwhite) were found to be at a higher risk for transplant-related mortality (TRM) compared to those with <3 (57% vs. 21% at 1 year, $p = 0.02$). RNA sequencing analysis was performed in pretransplantation samples and revealed the upregulation of various complement pathways in HSCT-TMA patients who had gene variants in comparison to those without variants and HSCT-TMA. Furthermore, *CFHR3* and *CFHR1* were considered other pathogenic variants. CFHR1 and CFHR3 proteins are also part of the complement factor H-related protein family and act as complement regulators. Mutations to these genes have been related to atypical hemolytic uremic syndrome, which is also a complement-mediated disorder [56,57]. In Figure 1, the role of potential gene variants of alternative complement pathway genes in the pathogenesis of HSCT-TMA is presented.

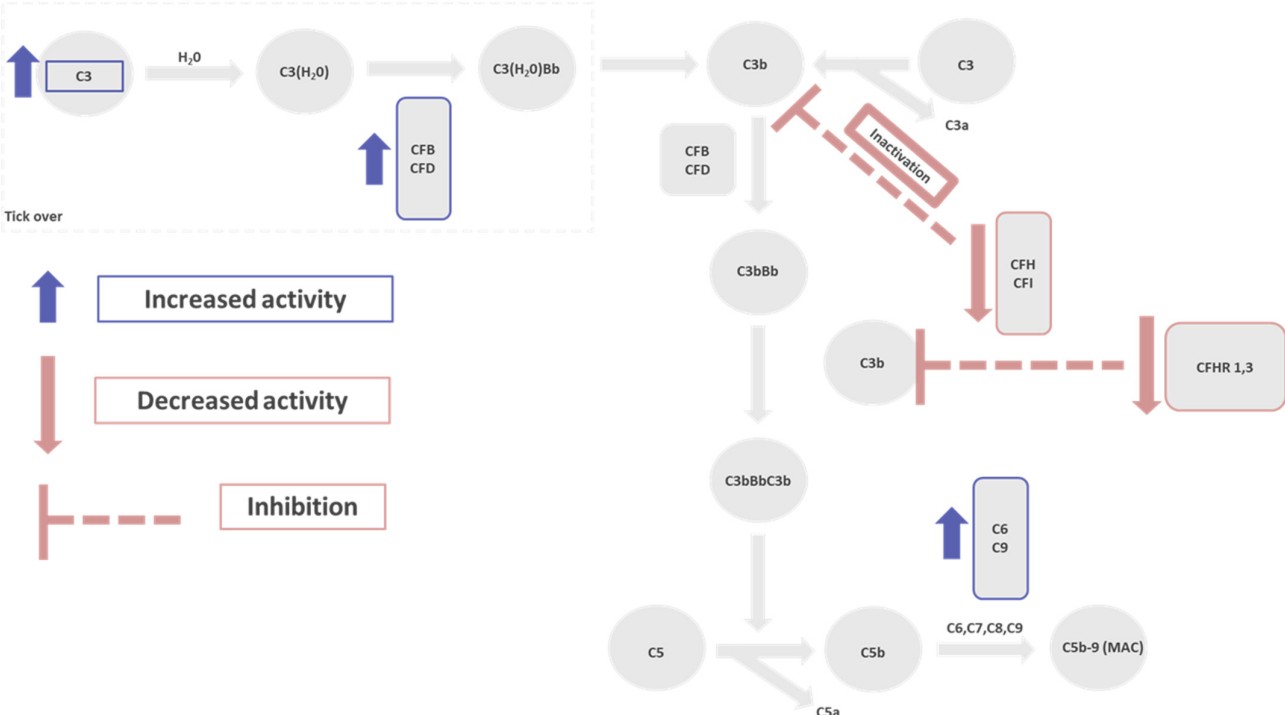

**Figure 1.** Role of potential gene variants of alternative complement pathway genes in the pathogenesis of HSCT-TMA. HSCT-TMA: hematopoietic stem cell transplantation-associated thrombotic microangiopathy; MAC; membrane attack complex.

In a cohort of six pediatric patients with HSCT-TMA, five of the six patients (83%) had heterozygous *CFHR3-CFHR1* gene deletion [58]. Nozawa et al. reported a case report of a 1-year-old girl who underwent autologous HSCT for neuroblastoma and developed HSCT-TMA [59]. After targeted genotyping in aHUS genes, a heterozygous *CFHR3-CFHR1* gene deletion was found, leading to excess complement. Ardissino et al., in their study, performed targeted next-generation sequencing (NGS) in both recipients and their donors and found complement-regulatory-related mutations in 6/16 donors of the patients who developed HSCT-TMA [60]. Variants in *CFB* were reported in a donor of a patient with HSCT-TMA. *CFB* encodes complement factor B, which is part of the alternative complement pathway and provides catalytic activity to the C3 and C5 convertases [61]. However, in this study, standardized criteria were not implemented for the diagnosis of HSCT-TMA [62]. In a previous study from our team, we studied 40 patients with HSCT-TMA, the donors of 18 patients who developed TMA, and 40 non-TMA HCT recipients [63]. NGS in TMA variants was performed in genomic DNA obtained from the pretransplant blood of the study's participants. We showed that in HSCT-TMA, there was a significantly higher frequency of variants in *ADAMTS13* ($p < 0.025$), *C3* ($p < 0.040$), *CFB* ($p < 0.002$), *CFH* ($p < 0.002$), and *CFI* ($p < 0.041$) in comparison to patients without TMA. Moreover, variants were identified in exonic/splicing/untranslated regions (UTRs) of *ADAMTS13* ($p < 0.047$), *C3* ($p < 0.003$), *CFH* ($p < 0.013$), and *CFI* ($p < 0.044$) in HSCT-TMA patients in comparison to those without. Zhang et al., in their study, showed that rare variants in the VWF clearance pathway were significantly associated with HSCT-TMA ($p = 0.008$) [64]. Furthermore, the *LRP1* variant was significantly increased in HSCT-TMA patients compared to controls ($p = 0.025$). LRP1 is a member of the low-density lipoprotein receptor family, implicated in VWF clearance and the protection against oxidative stress.

Rachakonda et al., in their study, showed that a rs3092936 single-nucleotide polymorphism (SNP) in the *CD40 ligand* (CD40L) gene was related to the development of HSCT-TMA [65]. Leimi et al. found that patients with endotheliopathies (HSCT-TMA, SOS/VOD, CLS) had variants in several complement pathways. The most notable were

detected in the terminal pathway (*C6* and *C9*), lectin pathway (*MASP1*), and *ITGAM*, which encodes CD11b, a part of the β-integrin complement receptor type 3, CR3 [66]. Genetic variants in the ITGAM gene have been related to systemic lupus erythematosus (SLE) and pre-eclampsia [67,68]. However, Okamura et al. performed NGS in 17 genes of the genome obtained from 30 HSCT recipients (15 with HSCT-TMA) and failed to identify an association between complement-associated genetic variants and HSCT-TMA [36]. To summarize, the presence of specific gene variants and SNPs, especially in complement-related genes, might act as a risk factor for HSCT-TMA development (first hit). However, data are lacking to make conclusions about whether the use of pre-transplantation genetic testing would be helpful for the identification of patients who are at great risk for HSCT-TMA development. To summarize, as reported in the published studies, 29 to 83% of patients with HSCT-TMA have one or more pathogenic genetic variant that predispose them to the development of this syndrome. We have to underline, as mentioned above, that other factors also contribute to the pathogenesis of HSCT-TMA, except genetic background. In Table 1, we present studies examining the role of gene variants as risk factors for the development of HSCT-TMA.

**Table 1.** Studies examining the role of gene variants as risk factors for the development of HSCT-TMA.

| First Author, Year | Study Design | Genetic Analysis | Number of Study Participants | Age Group of Patients | Genetic Variants Characterized as Pathogenic |
|---|---|---|---|---|---|
| Jodele 2013 [58] | Case series | NGS of alternative complement pathway genes (*CFH*, *CFI*, *MCP*, *CFB*, *CFHR1,3,5*) | 6 HSCT-TMA patients | Pediatric patients | *CFHR3*, *CFHR1* |
| Jodele 2016 [49] | Prospective study, case–control | NGS of 17 genes (*CFH*, *CFHR1*, *CFHR3*, *CFHR4*, *CFHR5*, *CD55*, *CD59*, *CD46*, *CFI*, *CFB*, *CFP*, *C5*, *ADAMTS13*, *CFD*, *C3*, *C4BPA*, and *THBD*) and RNA expression analysis | 90 alloHSCT recipients (34 with HSCT-TMA) | Pediatric patients | *CFHR5*, *CFI*, *ADAMTS13*, *CFHR3*, *CFHR1* |
| Ardissino 2017 [60] | Case series | NGS of specific genes (*CFH*, *CFHR1*, *CFHR3*, *CFHR4*, *CFHR5*, *CFI*, *CFB*, *CD46*, *C3*, *DGKe*, and *THBD*) | 16 HSCT-TMA patients and their donors | Adult and pediatric patients | *CFH*, *CFHR3*, *CFHR5*, *CFI*, *CFB*, *C3* |
| Nozawa 2018 [59] | Case report | NGS of aHUS genes (*CFH*, *CFHR1*, *CFHR3*, *CFHR4*, *CFHR5*, *CFI*, *CFB*, *CD46*, *C3*, and *THBD*) | 1 autologous HSCT recipient | 1 year-old patient | *CFHR3*, *CFHR1* |
| Rachakonda 2018 [65] | Prospective study | *CD40L* and *THBD* SNP genotyping | 966 HSCT recipients | Adult patients | rs3092936 SNP in *CD40L* |
| Gavriilaki 2020 [63] | Prospective study, case–control | NGS with a customized complement-related gene panel (*CFH*, *CFHR1*, *CFHR3*, *CFHR4*, *CFHR5*, *CFI*, *CFB*, *CFD*, *C3*, *CD55*, *C5*, *CD46*, *ADAMTS13*) | 80 HSCT recipients (40 with HSCT-TMA, 40 without TMA) | Adult patients | *ADAMTS13*, *C3*, *CFB*, *CFH*, *CFI* |
| Okamura 2021 [36] | Retrospective, case–control study | NGS in 17 genes (*C3*, *C5*, *CFB*, *CFD*, *CFH*, *CFI*, *CFP*, *C4BPA*, *CD46*, *CD55*, *CD59*, *THBD*, *CFHR1*, *CFHR3*, *CFHR4*, *CFHR5*, and *ADAMTS13*) | 30 HSCT recipients (15 with HSCT-TMA, 15 without) | Adult patients | No association was found |
| Zhang 2023 [64] | Prospective study, case–control | NGS in panel of 52 genes, focusing on 5 pathways: complement system, vWF function and associated proteins, vWF clearance, ADAMTS13 function and associated proteins, and endothelial activation | 198 HSCT recipients (100 with HSC-TMA, 98 without TMA) | Adult patients | Variants in the vWF clearance pathway, *LRP1* |

**Table 1.** *Cont.*

| First Author, Year | Study Design | Genetic Analysis | Number of Study Participants | Age Group of Patients | Genetic Variants Characterized as Pathogenic |
|---|---|---|---|---|---|
| Leimi 2023, [66] | Retrospective, case-control study | WES in variations and mutations in classical, lectin, or terminal pathway factors or in the membrane-bound components of the complement system | 109 HSCT patients (17 with endotheliopathy, including HSCT-TMA, SOS/VOD, capillary leak syndrome) | Pediatric patients | *C6*, *C9*, *MASP1*, *ITGAM* |

HSCT-TMA: hematopoietic stem cell transplantation-associated thrombotic microangiopathy; NGS: next-generation sequencing; HSCT: hematopoietic stem cell transplantation; SNP: single-nucleotide polymorphism; vWF: von Willebrand factor; WES: whole-exome sequencing; SOS/VOD: veno-occlusive disease/sinusoidal obstruction syndrome.

## 3. SOS/VOD and Genetic Susceptibility

SOS/VOD is a rare and severe alloHSCT complication, characterized by high mortality (up to 80%) [69]. Chemotherapy and radiotherapy (used as a preconditioning regimen or for the initial treatment of the hematological malignancy) result in endothelial cell injury and activation and the release of inflammatory and procoagulant factors [70]. In these factors, vWF and tissue factor (TF) are included, leading to the formulation of microvascular clots in sinusoids and venules of the liver, obstruction to the blood flow, and portal hypertension [71]. Hyperbilirubinemia, painful hepatomegaly, jaundice, rapid weight gain unresponsive to diuretics, and ascites are among the clinical manifestations of SOS/VOD [72]. Multiple organ failure affecting the lungs (hepatopulmonary syndrome), kidneys (hepatorenal syndrome), and central nervous system (CNS) has also been described [73]. In the guidelines of the British Society of Hematology, ursodeoxycholic acid is recommended for the prevention of SOS/VOD post-HSCT [74]. Moreover, defibrotide, an agent with protective properties for endothelial cells, has been investigated as a preventive agent for this syndrome and is approved for the management of patients with moderate/severe SOS/VOD [75].

Bucalossi et al. studied seven patients who underwent alloHSCT (three with SOS/VOD) and detected two *CFH* variants in patients with SOS/VOD [76]. The methylenetetrahydrofolate reductase (*MTHFR*) C677T/A1298C genotype has been found as a possible risk factor for the development of SOS/VOD [77]. MTHF is a key enzyme in regulating folate and homocysteine metabolism [78]. Moreover, in children with rs4693608 and rs4364254 SNPs in the *heparinase* gene, the incidence of SOS/VOD syndrome decreased [79]. Various SNPs in the *glutathione S-transferase* (GST) gene, encoding the GST enzyme, essential for the metabolism of busulfan (which used a conditioning regimen), have been associated with SOS/VOD disease post-HSCT [80]. In Table 2, the genes implicated in the pathogenesis of various endothelial injury syndromes are summarized.

**Table 2.** Genes implicated in the pathogenesis of endothelial injury post-HSCT.

| Gene | Role of Gene Product | Endothelial Injury Syndrome |
|---|---|---|
| *CFHR1* | The protein encoded by this gene belongs to the complement factor H protein family, acting as a complement regulatory protein. | HSCT-TMA |
| *CFHR3* | The protein encoded by this gene belongs to the complement factor H protein family, acting as a complement regulatory protein. | HSCT-TMA |
| *CFHR5* | The protein encoded by this gene belongs to the complement factor H protein family, acting as a complement regulatory protein. | HSCT-TMA |

**Table 2.** *Cont.*

| Gene | Role of Gene Product | Endothelial Injury Syndrome |
|------|---------------------|----------------------------|
| *CFH* | The gene product acts as a regulator of complement activation. | SOS/VOD |
| *CFI* | CFI protein regulates complement activation by cleaving C3b and C4b. | HSCT-TMA |
| *CFB* | The active subunit Bb of CFB (activated by factor D) is a protease that binds with C3b to form the alternative pathway, C3 convertase. | HSCT-TMA |
| *C3* | C3 has a crucial role in the activation of the complement system. | HSCT-TMA |
| *MASP1* | This gene encodes a protein involved in the lectin pathway of the complement system. | HSCT-TMA |
| *C6* | C6 is part of the complement system and is involved in the formulation of MAC. | HSCT-TMA |
| *C9* | C9 is part of the complement system and is involved in the formulation of MAC. | HSCT-TMA |
| *ITGAM* | It encodes a protein subunit that forms the heterodimeric complement receptor 3. | HSCT-TMA |
| *CD40L* | CD40 is a transmembrane glycoprotein primarily expressed on activated CD4+ T cells, playing a major role in the regulation of immune response. | HSCT-TMA |
| *LRP1* | This receptor takes part in several cellular processes, including the clearance of apoptotic cells. | HSCT-TMA |
| *ADAMTS13* | A metalloprotease enzyme that takes part in vWF cleavage. | HSCT-TMA |
| *MTHFR* | A key enzyme that regulates the folate and homocysteine metabolism. | SOS/VOD |
| *Heparinase* | Heparinase is an enzyme that plays a major role in the inflammatory process. | SOS/VOD |
| *GST* | Specific genetic polymorphisms have an impact on the busulfan pharmacokinetics. | SOS/VOD |

HSCT: hematopoietic stem cell transplantation; HSCT-TMA: hematopoietic stem cell transplantation-associated thrombotic microangiopathy; CFHR: complement factor H related; CFH: complement factor; SOS/VOD: veno-occlusive disease/sinusoidal obstruction syndrome, CFI: complement factor I; C3b: complement component 3b; C4b: complement component 4b; CFB: complement factor B; C3: complement component 3; MASP1: mannan-binding lectin serine protease 1; C6: complement component 6; MAC: membrane attack complex; C9: complement component 9; ITGAM: integrin alpha M; CD40L: CD40 ligand; LRP1: LDL receptor-related protein 1; ADAMTS13: a disintegrin and metalloproteinase with a thrombospondin type 1 motif, member 13; vWF: von Willebrand factor; MTHFR: methylenetetrahydrofolate reductase; GST: glutathione S-transferase.

## 4. CRS and ICANS as Endotheliopathies: The Genetic Background

CRS and ICANS are the two principal complications post-CAR-T therapy that indicate an unmet need for urgent patient management [81,82]. The frequency of the two syndromes presented in phase 3 trials varies between the different patient cohorts (CRS 37–93% and ICANS 23–65%) [83–85]. Clinical presentations of CRS include multiple system symptoms, from general symptoms such as fever, fatigue, and anorexia to specific organ damage symptoms (respiratory failure, acute kidney injury, hypotension, arrhythmias, acute heart failure, liver damage, and GI symptoms) [86]. ICANS manifestations are disparate among patients and include headache, the disruption of consciousness level, encephalopathy, and seizures [87]. The pathophysiology of neurotoxicity and CRS is not fully understood yet, but studies have supported the concept that the main mechanism involved is endothelial injury [88,89].

Several markers of endothelial injury have been found to increase in patients with CRS and ICANS. Angiopoietin-II and vWF, as markers of endothelial injury, were found to be elevated in patients with severe ICANS (stage 4) [90]. An increased angiopoietin-II/angiopoietin-I ratio has been observed in those who were diagnosed with severe ICANS (grade $\geq$ 3) compared to those with grades 0 to 2 [91]. The cytokine production sequence of the CRS results in the activation of vascular inflammation leading to abnormal function of the endothelium [24]. In particular, the EASIX and the modified EASIX (m-EASIX), as markers involved in endothelial injury, have been suggested as potential predictors of severe ICANS and CRS [10,92]. The activation of endothelial cells in the CNS and of pericytes followed by CAR-T immunotherapy leads to the production of interleukin-6 (IL-6), vascular endothelial growth factor (VEGF), and interferon-$\gamma$ (IFN-$\gamma$), increasing the permeability of the blood–brain barrier (BBB) [93–96]. The increase in these molecules combined with increased levels of tumor necrosis factor-alpha (TNF-a) (secreted by activated CAR-T cells) activates the production of matrix metalloproteinase (MMP) 2 and 9 by CNS endothelial cells [97,98]. MMPs contribute to endothelial damage, activating cell adhesion molecules and further increasing the permeability of the BBB [99].

The detection of the genetic mutations contributing to the onset of cytokine release after CAR-T immunotherapy could lead to the development of preventive measures for CRS [26,89]. The *PFKFB4* gene mediates the development of CRS in patients who receive CD22 CAR-T immunotherapy. The upregulation of *PFKFB4* activates the glycolytic pathway, leading to cytokine production. Specifically, CRS is positively correlated with the presence of the *PFKFB4* gene, while the *PFKFB4* gene was gradually upregulated as CRS became more severe [26]. Wang et al. performed a DNA-sequencing analysis in patients who received CD19/CD22 CAR-T immune therapy. The study showed that a mutation in the *CX3CR1* gene (Mut I249/M280) leads to the onset of large B-cell lymphoma and to the presence of mild-severity CRS in patients who received CAR-T products [100]. In a clinical trial by Talleur et al., the *CRLF2-r*‡ hyperdiploid mutation was presented in patients with severe CRS after CAR-T therapy [101]. The PPM1D mutation (amplification+ overexpression in 17q chromosome) was presented in patients with severe ICANS (grade $\geq$ 3) in a cohort of 85 patients who received CAR-T immunotherapies for hematological malignancies [102]. Severe ICANS (grade $\geq$ 3) has been associated with the *DNMT3A*, *TE2*, and *ASXL1* gene mutations (DTA mutations). The presence of DTA mutations increased the risk of severe ICANS (58.9% vs. 25% in patients with the absence of DTA mutations, $p$ = 0.02) in a cohort of 114 patients with B-cell hematological malignancies [103]. In Figure 2, an overview of the principal pathogenetic mechanisms involved in the onset of toxicities after CAR-T immunotherapy is presented.

Zhou et al. developed an animal leukemic model and administered CD19 CAR-T immune therapy with short hairpin RNA (shRNA), targeting the IL-6 gene. The IL-6 downregulation helped in the CRS prevention in this model, suggesting that the interplay between IL-6 and IL-6 receptor (IL-6R) contributes to the onset of CRS [104]. The efficacy of tocilizumab, an IL-6R antagonist, in preventing CRS underlines that IL-6 plays an important role in CRS onset [105]. Interleukin-15 (IL-15) mediates the onset of neurotoxicity and CRS in patients who received CAR-T immunotherapy [106]. Zhang et al. conducted a study in a mouse model of CD-19 CAR-T immunotherapy along with increased IL-15 overexpression. One of the groups studied received IL-15a receptor (IL-15Ra). The overexpression of IL-15Ra reduced the severity and frequency of the CRS onset. These findings suggested that IL-15Ra can contribute to the prevention of CRS [107].

Anti-CD19 CAR-T immunotherapy with IL-6 gene downregulation can limit the adverse immunotoxicity originating from CAR-T immunotherapy [108]. The inhibition of nuclear factor kappa beta (NF-$\kappa\beta$) by the suspension of the two genes responsible for the transcription of this molecule (IRAK1, TRAF6) leads to the suppression of inflammation in vivo. This study reports the role of the IRAK1 and TRAF6 genes in the onset of CRS [109]. Zhang and colleagues conducted an in vitro study evaluating the efficacy of double-IL-6/INF-$\gamma$ knockdown for the prevention of CRS after anti-CD19 CAR-T therapy. The

study concluded that the double-gene knockdown contributes to the inhibition of CRS development in vitro compared to controls ($p < 0.001$) [110]. The granulocyte-macrophage colony-stimulating factor (GM-CSF) plays a vital role in the clinical manifestations of CRS and ICANS [111]. The knockout of genes interplaying in the transcription of GM-CSF reduced the inflammation mediated by cytokine release in patients who received CAR-T immunotherapies [112–116]. Guercio et al. conducted a study in vivo and administered the inducible caspase 9 (iC9) gene in a model of CAR-T cell lines. The induction of iC9 prevented cytokine production [117].

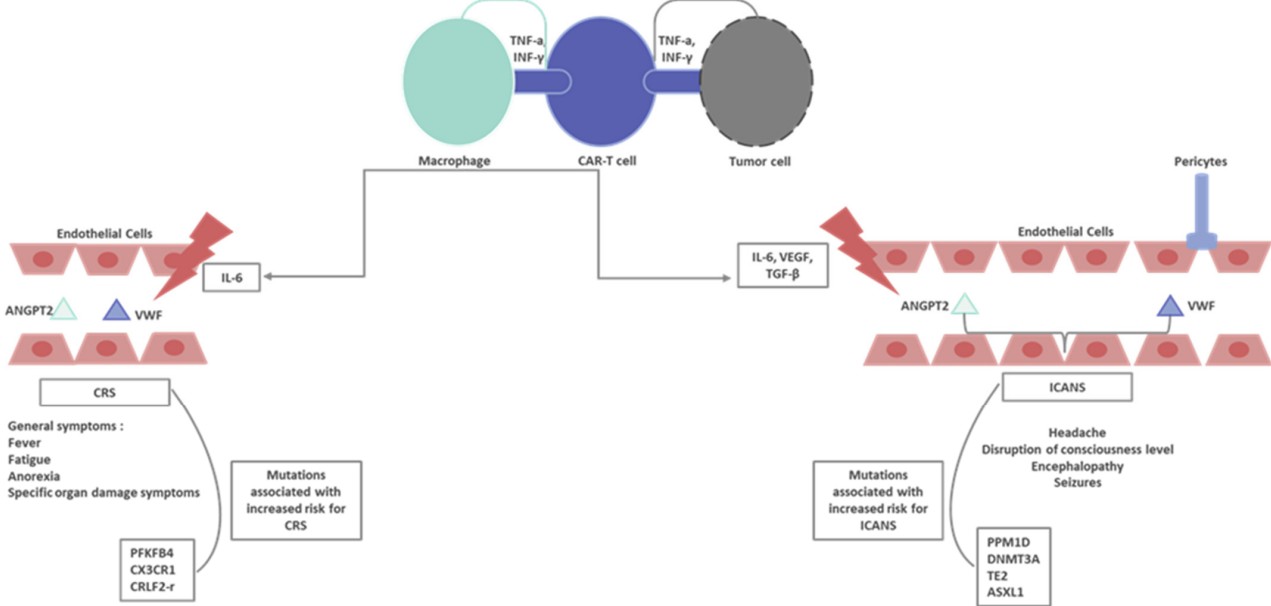

**Figure 2.** Overview of principal pathogenetic mechanisms involved in the onset of toxicities after CAR-T immunotherapy. TNF-a: tumor necrosis factor-alpha; INF-γ: interferon-γ; CAR-T: chimeric antigen receptor-T; IL-6: interleukin-6; ANGPT2: angiopoietin 2; VWF: von Willebrand factor; CRS: cytokine release syndrome; VEGF: vascular endothelial growth factor; TGF-β: transforming growth factor beta; ICANS: immune effector cell-associated neurotoxicity syndrome.

The *estrogen receptor-binding fragment-associated antigen 9 (EBAG9)* gene upregulation decreases the cytokine levels and can potentially prevent CRS onset [118]. Wirgers et al. demonstrated that *EBAG9* gene silencing in an animal model that received anti-CD8 CAR-T immunotherapy was not effective in the prevention of CRS [119]. The transcription suspension of cyclin-dependent kinase 7 (CDK7) inhibited the inflammatory release of cytokines and prevented CRS [120]. This study reports that the *CDK7* gene contributes to the onset of CRS after CAR-T therapies. Genetic susceptibility to CRS has been investigated in other clinical entities, such as COVID-19. Yang et al. studied the Gene Expression Omnibus database of patients with COVID-19 and CRS. The study concluded that the upregulation of *IL-6R, Toll-Like Receptor 4 (TLR4), Toll-Like Receptor 2 (TLR2)*, and *IFN-γ* genes can contribute to the onset of CRS in 10 patients with CRS and SARS-CoV-2 infection [121].

## 5. Conclusions

HSCT is a crucial and sometimes curative treatment approach for various hematological malignancies [122]. COVID-19 disease formed a challenge for both transplant physicians and their vulnerable patients [123]. Endothelial dysfunction, complement dysregulation, and the activation of coagulation cascade are implemented in the pathogenesis of various HSCT complications, which can be described as endotheliopathies. Variants in complement-related genes, such as *CFB, CFI, CFH-related, CFB, MASP-1*, and *C3*, lead to a predisposition to complement dysregulation. Classical, lectin, and alternative pathways are implicated. It remains unclear whether pre-transplantation genetic testing for the identification of these

variants in all the patients who are going to undergo an HSCT would be beneficial. Moreover, future studies should examine whether the patients with these variants might benefit from complement inhibitors. Notably, complement inhibitors, and the first-ever used in HSCT-TMA, eculizumab, increased the response rates of these patients. Furthermore, more studies are essential for a better understanding of the genetic base of SOS/VOD, focusing on both coagulation and complement-related genes.

CAR-T cell immunotherapy brought a revolution in the management of patients with refractory/relapsed b-cell hematological malignancies. CRS and ICANS, novel toxicities that have arisen from the use of CAR-T cell products in everyday clinical practice, reduce the therapeutic efficacy of this treatment approach. More studies investigating the genetic susceptibility to the development of severe CRS and ICANS after CAR-T infusions are essential. Genes implicated in the pathogenesis of CRS accompanying other clinical entities such as COVID-19 can be also examined for this purpose. This approach might be helpful in both the development of ambulatory CAR-T infusion programs and the identification of patients who could benefit from prophylactic measures, such as the administration of tocilizumab as a prophylactic agent. Artificial intelligence algorithms that recognize patients who are at greater risk for severe complications after the CAR-T infusion can include the genetic data of patients in the future [124].

**Author Contributions:** Conceptualization, E.G. and P.E.; methodology, P.K.; investigation, N.E; writing—original draft preparation, P.E. and N.E.; writing—review and editing, E.G., A.T., and M.K. visualization, E.G. and P.E.; supervision, E.G.; project administration, E.G. All authors have read and agreed to the published version of the manuscript.

**Funding:** This research received no external funding.

**Conflicts of Interest:** The authors declare no conflicts of interest.

## Abbreviations

| | |
|---|---|
| ADAMTS13 | A disintegrin and metalloproteinase with a thrombospondin type 1 motif, member 13 |
| aGVHD | Acute graft-versus-host disease |
| alloHSCT | Allogeneic HSCT |
| BBB | Blood–brain barrier |
| C3b | Complement component 3 b |
| CAR-T | Chimeric antigen receptor-T |
| CD40L | CD40 ligand |
| CDK7 | Cyclin-dependent kinase 7 |
| CFH | Complement factor H |
| CFI | Complement factor I |
| CLS | Capillary leak syndrome |
| CNIs | Calcineurin inhibitors |
| CNS | Central nervous system |
| COVID-19 | Coronavirus disease 2019 |
| CRS | Cytokine release syndrome |
| CVD | Cardiovascular disease |
| EASIX | Endothelial Activation and Stress Index |
| EBAG9 | Estrogen receptor-binding fragment-associated antigen 9 |
| GI | Gastrointestinal |
| GM-CSF | Granulocyte macrophage colony-stimulating factor |
| GST | Glutathione S-transferase |
| GvHD | Graft-versus-host disease |
| HLA | Human leukocyte antigen |
| HSCT | Hematopoietic stem cell transplantation |
| HSCT-TMA | HSCT-associated thrombotic microangiopathy |
| iC9 | Inducible caspase 9 |

| | |
|---|---|
| ICANS | Immune effector cell-associated neurotoxicity syndrome |
| IFN-γ | Interferon-γ |
| IL-15 | Interleukin-15 |
| IL-15Ra | IL-15a receptor |
| IL-6 | Interleukin-6 |
| IL-6R | IL-6 receptor |
| MAC | Membrane attack complex |
| m-EASIX | Modified EASIX |
| MMP | Matrix metalloproteinase |
| MTHFR | Methylenetetrahydrofolate reductase |
| NF-κβ | Nuclear factor kappa beta |
| shRNA | Short hairpin RNA |
| SLE | Systemic lupus erythematosus |
| SNP | Single nucleotide polymorphism |
| SOS/VOD | Veno-occlusive disease/sinusoidal obstruction syndrome |
| TF | Tissue factor |
| TLR2 | Toll Like Receptor 2 |
| TLR4 | Toll Like Receptor 4 |
| TNF-a | Tumor necrosis factor alpha |
| TRM | Transplant-related mortality |
| UTR | Untranslated region |
| VEGF | Vascular endothelial growth factor |
| vWF | Von Willebrand factor |

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
