# Peer review of "Genetic Susceptibility in Endothelial Injury Syndromes after Hematopoietic Cell Transplantation and Other Cellular Therapies: Climbing a Steep Hill"

_cimb, doi:10.3390/cimb46050288_

Round 1

Reviewer 1 Report

Comments and Suggestions for Authors

Allogeneic hematopoietic stem cell transplantation (allo-HSCT) holds the potential to provide a cure for various malignant and non-malignant diseases affecting the lymphohematopoietic system. Despite significant advancements since the initial breakthroughs in Seattle over 50 years ago, non-relapse mortality continues to be a significant challenge in allo-HSCT. There is a growing body of evidence suggesting that endothelial dysfunction plays a role in several serious complications of allo-HSCT, including SOS/VOD, HSCT-TMA, and refractory acute GVHD. This review provides a comprehensive analysis of the endothelium involvement in severe complications following alloSCT. It also discusses the ongoing efforts to identify genes that can predict these complications, such as indicators of endothelial vulnerability and markers of endothelial injury. In this paper, the authors discuss the implications of our current understanding of transplant-associated endothelial pathology for the prevention and management of complications after allo-HSCT. However, there are a few questions that need clarification for the readers.

1.     Genetic susceptibility is acknowledged as a predisposing risk factor in the development of endothelial injury syndrome. Authors could enhance readability by including a table that outlines the correlation between endothelial function-related genes and various endothelial injury syndromes. 

2.     2. Given that most CAR-T therapies utilize autologous T-cells, it is important to consider the distinction between allo-HCT for donor origin.

Comments on the Quality of English Language

The quality of the English language is satisfactory.

Author Response

Answer to Reviewer’s 1 Comments

Dear Reviewer 1,

Allogeneic hematopoietic stem cell transplantation (allo-HSCT) holds the potential to provide a cure for various malignant and non-malignant diseases affecting the lymphohematopoietic system. Despite significant advancements since the initial breakthroughs in Seattle over 50 years ago, non-relapse mortality continues to be a significant challenge in allo-HSCT. There is a growing body of evidence suggesting that endothelial dysfunction plays a role in several serious complications of allo-HSCT, including SOS/VOD, HSCT-TMA, and refractory acute GVHD. This review provides a comprehensive analysis of the endothelium involvement in severe complications following alloSCT. It also discusses the ongoing efforts to identify genes that can predict these complications, such as indicators of endothelial vulnerability and markers of endothelial injury. In this paper, the authors discuss the implications of our current understanding of transplant-associated endothelial pathology for the prevention and management of complications after allo-HSCT.

Answer: We would like to deeply thank the reviewer for the time dedicated to reviewing our manuscript. The reviewer’s comments and suggestions were substantial for the clarity and quality of our work. Thanks once again.

However, there are a few questions that need clarification for the readers.

  1. Genetic susceptibility is acknowledged as a predisposing risk factor in the development of endothelial injury syndrome. Authors could enhance readability by including a table that outlines the correlation between endothelial function-related genes and various endothelial injury syndromes.

Answer: Thanks for this idea. We included a table summarizing the genes implicated in endothelial injury syndromes after the HSCT.

  1. 2. Given that most CAR-T therapies utilize autologous T-cells, it is important to consider the distinction between allo-HCT for donor origin.

Answer: What an excellent idea! Thank you for this comment. We rephrased our sentence with the following: “Several markers of endothelial injury have been found to increase in patients with CRS and ICANS.”

Reviewer 2 Report

Comments and Suggestions for Authors

An interesting and well written review.

General comment - the authors show associations between endothelial syndromes in HSCT and CRS in CAR-T therapy, and specific genetic associations. Can they give an estimate as to how many patients with these symptoms have a genetic association that they describe - and why do most patients not have that association?

Minor comments:
Line 38: - should 'morality' be morbidity?
Line 138: - 'in comparison to 9% of those without' - ?without what?

lines 143-146: - this does not make sense - please re-write

Line 260 - is not the PFKFB4 gene found in everyone - not just those with grade 4 CRS?

Lines 283-284 - IL-6 interacts with IL-6??

Comments on the Quality of English Language

Very good, minor editing required

Author Response

Answer to Reviewer’s 2 comments 

Dear Reviewer, 

We would like to thank the reviewer for the time dedicated to reviewing our work. 

An interesting and well written review.

Answer: Thank you for this comment 

General comment - the authors show associations between endothelial syndromes in HSCT and CRS in CAR-T therapy and specific genetic associations. Can they give an estimate as to how many patients with these symptoms have a genetic association that they describe - and why do most patients not have that association?

Answer: Thanks for these suggestions. We added the following sentences at the end of section 2 as you suggested: “To summarize, as reported in the published studies 29 to 83% of patients with HSCT-TMA have one or more pathogenic genetic variants that predispose to the development of this syndrome. We have to underline, as mentioned above, that also other factors contribute to the pathogenesis of HSCT-TMA, except genetic background.”

Minor comments:

Line 38: - should 'morality' be morbidity?

Answer: We would like to thank the reviewer for this correction. That is true. Sorry for this missense.

Line 138: - 'in comparison to 9% of those without' - ?without what?

Answer: Thanks for this observation. We added the phrase: “without TMA after the transplantation”. 

lines 143-146: - this does not make sense - please re-write

Answer: We would like to thank the reviewer for this valuable comment. Indeed re-writing was necessary to clarify the meaning of this sentence. We rephrased the sentence in the following way: “RNA sequencing analysis was performed in pretransplantation samples and revealed upregulation of various complement pathways in HSCT-TMA patients who had gene variants in comparison to those without variants and HSCT-TMA.”

Line 260 - is not the PFKFB4 gene found in everyone - not just those with grade 4 CRS?

Answer: Thanks for this comment. The presence of the PFKFB4 gene was associated with the development of CRS in CAR-T immunotherapy recipients. However, higher upregulation of PFKFB4 is observed in patients with severe CRS (PMID: 35902861). 

Lines 283-284 - IL-6 interacts with IL-6??

Answer: We would like to thank you for this correction. IL-6 interacts with the receptor of IL-6 (IL-6R). 
